# Increased Azithromycin Susceptibility of Multidrug-Resistant Gram-Negative Bacteria on RPMI-1640 Agar Assessed by Disk Diffusion Testing

**DOI:** 10.3390/antibiotics9050218

**Published:** 2020-04-29

**Authors:** Milton Meerwein, Andrea Tarnutzer, Michelle Böni, Françoise Van Bambeke, Michael Hombach, Annelies S. Zinkernagel

**Affiliations:** 1Department of Infectious Diseases and Hospital Epidemiology, University of Zurich, 8091 Zurich, Switzerland; 2Pharmacologie cellulaire et moléculaire, Louvain Drug Research Institute, Université catholique de Louvain, B-1200 Brussels, Belgium; 3Institute of Medical Microbiology, University of Zurich, 8091 Zurich, Switzerland

**Keywords:** azithromycin, Gram-negative, *Pseudomonas aeruginosa*, Enterobacteriaceae, susceptibility testing, MIC, disk diffusion, multidrug-resistant

## Abstract

Increasing antibiotic resistances and a lack of new antibiotics render the treatment of Gram-negative bacterial infections increasingly difficult. Therefore, additional approaches are being investigated. Macrolides are not routinely used against Gram-negative bacteria due to lack of evidence of in vitro effectiveness. However, it has been shown that Pseudomonas spp. are susceptible to macrolides in liquid RPMI-1640 and clinical data suggest improvement in patients’ outcomes. So far, these findings have been hardly applicable to the clinical setting due to lack of routine low-complexity antimicrobial susceptibility testing (AST) for macrolides. We therefore optimized and compared broth microdilution and disk diffusion AST. Multidrug-resistant Gram-negative bacteria (*Escherichia coli, Enterobacter cloacae, Klebsiella pneumoniae, Pseudomonas aeruginosa*) were tested for azithromycin susceptibility by disk diffusion and broth microdilution in Mueller–Hinton and RPMI-1640 media. Azithromycin susceptibility of Enterobacteriaceae and a subgroup of *P. aeruginosa* increased significantly on RPMI-1640 agar compared to Mueller–Hinton agar. Further, a significant correlation (Kendall, τ, *p*) of zone diameters and minimal inhibitory concentrations (MICs) was found on RPMI-1640 agar for *E. coli* (−0.4279, 0.0051), *E. cloacae* (−0.3783, 0.0237) and *P. aeruginosa* (−0.6477, <0.0001). Performing routine disk diffusion AST on RPMI-1640 agar may lead to the identification of additional therapeutic possibilities for multidrug-resistant bacterial infections in the routine clinical diagnostic setting.

## 1. Introduction

Increasing bacterial resistance to antibiotics and a lack of new antimicrobial compounds render the treatment of bacterial infections more and more difficult [1]. One example is the rise of pathogens carrying multidrug resistance (MDR) plasmids that confer resistance even against last-resort antibiotics such as carbapenems and colistin. Treatment options for such MDR isolates are extremely limited and the development of new drugs has almost come to a halt over the last two decades [2]. As an alternative approach, old drugs such as fosfomycin or nitrofurantoin, which display a favorable epidemiologic susceptibility profile due to infrequent use and concomitant low selection pressure, have been revived. Another approach would be to reconsider susceptibility testing methods for their actual ability to predict the in vivo effectiveness of established drugs against bacterial groups that are currently considered naturally resistant. An example are macrolides which are considered to be active against Gram-positive infections; in clinics they are often used in patients with acute lower respiratory tract infections [3]. Resistances against macrolides in Gram-positive bacteria are often due to methylation or mutations in domain V of the 23S rRNA gene, which is targeted by macrolides, or active efflux [4]. In contrast, most Gram-negative bacteria are considered intrinsically resistant to macrolides, as their outer membrane is highly impermeable to hydrophobic substances and possesses very efficient efflux pump systems [5,6]. While macrolides possess good activity against a few distinct Gram-negative microorganisms, such as Bartonella spp. or Campylobacter spp., they are less effective in *Enterobacteriaceae* [7,8]. Nevertheless, clinical trials suggest that macrolides could improve the clinical outcome of patients with Gram-negative infections [9,10]. A meta-analysis showed an improved clinical outcome in cystic fibrosis patients routinely treated with macrolides, an effect attributed to immunomodulatory effects [11].

Antimicrobial susceptibility testing (AST) follows Kirby–Bauer’s original disk diffusion technique by growing bacteria on cation-adjusted Mueller–Hinton (MH) agar plates, a comparably rich growth medium ensuring sufficient bacterial growth and reproducible results [12]. As azithromycin is not a standard treatment option for Gram-negative infections, susceptibility data and clinical breakpoints have not been published by the European Committee on Antimicrobial Susceptibility Testing (EUCAST) [13]. In contrast with the other macrolides, azithromycin has been shown to be more basic in character, which results in higher permeability across the outer membrane [7]. Thus, recently, a putative minimal inhibitory concentration (MIC) epidemiological cut-off value (ECOFF) of ≥32 mg/L was suggested for *Escherichia coli* using agar dilution according to the Clinical and Laboratory Standards Institute (CLSI), in order to discriminate wild-type from increased efflux or decreased permeability isolates, and to provide a decision rationale for alternative treatment of diarrheal infections [14].

Bacterial growth media have not been modelled to reflect the physiological situation in the patient but rather to ensure sufficient in vitro growth and reproducibility. Thus, AST in human serum or lung fluids would be ideal but is not feasible in a routine setting. Eukaryotic growth medium has been proposed as an alternative. Recent studies showed that azithromycin MICs were lower in the eukaryotic growth medium Roswell Park Memorial Institute 1640 (RPMI-1640) [15,16,17] or Dulbecco Modified Eagle Medium (DMEM) as compared to MH medium [18]. Thus, modifying in vitro AST parameters in order to more closely mirror in vivo conditions may reveal additional treatment options for Gram-negative MDR infections [19]. The goal of this study was, hence, to transfer those observations into a routine clinical microbiology setting. As in diagnostics, antibiotic susceptibility is mostly assessed by agar-based disk diffusion testing; we evaluated a quick and simple method for AST by developing a disk diffusion assay on RPMI-1640 agar. This allows to routinely assess the antimicrobial activity of azithromycin against various Gram-negative bacteria.

## 2. Results

### 2.1. *Enterobacteriaceae* Showed Enhanced Azithromycin Susceptibility on RPMI-1640 Agar

Median azithromycin inhibition zones of *Enterobacteriaceae* (*E. coli, K. pneumoniae* and *E. cloacae*) were significantly larger on RPMI-1640 agar plates as compared to standard MH plates (median diameter increase of 7–8 mm (Table 1)).

The MIC-colored zone method, as described by Kronvall et al., which we adopted and which is used by EUCAST [20], showed significant correlation of MICs and inhibition zone diameters (Figure 1) in RPMI-1640 medium for *E. coli* and *E. cloacae* (Kendall’s τ =−0.4279, −0.3783, *p* = 0.0051, 0.0237, respectively). In *K. pneumoniae* the correlation was not statistically significant although a trend was detectable.

The inhibition zone diameter of every individual *Enterobacteriaceae* isolate increased on RPMI-1640 agar as compared to MH agar, and for *E. coli* a putative split into a more susceptible and a resistant population seemed to occur, although corresponding numbers are low (Figure 1). Of note, clinical carbapenemase-positive *Enterobacteriaceae* strains consistently showed larger azithromycin inhibition zones on RPMI-1640 than on MH agar (Appendix A). Inhibition zones for the other antibiotics tested (cefepime, tobramycin, clarithromycin, erythromycin, tetracycline and tigecycline) did not differ between RPMI-1640 and MH agar in any of the examined species. In contrast to our freshly prepared RPMI-1640 agar plates, the commercially available RPMI-1640 agar plates routinely used for mycology assays were not suitable due to inhibition of bacterial growth. To exclude the possibility that the increased susceptibility found in RPMI-1640 was exclusively due to the added sodium bicarbonate, we assessed the azithromycin MICs in MH with sodium bicarbonate in a set of four strains of each species. Consistent with the data of Ersoy et al. [18], the MICs towards azithromycin were lower in MH with sodium bicarbonate compared to MH without sodium bicarbonate. However, the tested isolates remained less susceptible to azithromycin in MH with sodium bicarbonate as compared to RPMI-1640 (Appendix A). Thus, we conclude that sodium bicarbonate is not the only component in RPMI-1640 leading to lower resistance towards azithromycin.

### 2.2. Azithromycin Susceptibility Increased on RPMI-1640 Agar in a Subgroup of *P. aeruginosa*

In the 55 clinical *P. aeruginosa* isolates, the average azithromycin inhibition zone diameters increased significantly on RPMI-1640 agar as compared to MH agar, although to a lower extent than in the Enterobactericeae strains (Table 1). This was mainly due to a subgroup of strains displaying inhibition zone diameters ranging from 6 to 12 mm with minimal difference between RPMI-1640 and MH agar. In contrast, two other delimited sub-populations showed a trend towards higher azithromycin susceptibility on RPMI-1640 as compared to MH agar, one group ranging from 13 to 20 mm and the other group displaying large inhibition zone diameters (24 to 30 mm) on RPMI-1640 agar (Figure 2).

Nineteen out of the 29 clinical *P. aeruginosa* isolates from the European study batch derived from cystic fibrosis patients. Ten of those isolates harbored a mutation in the gene coding for Domain V of the 23S ribosomal subunit, which is known to confer azithromycin resistance [21]. All the isolates containing a mutation presented with small inhibition zone diameters ranging from 6–8 mm and MICs ≥ 128 mg/L in both media, while the rest of the CF patients’ isolates showed increased susceptibility in RPMI-1640 medium (Figure 3). None of the hospital-acquired isolates (*n* = 10) carried a mutation. Testing the 26 clinical isolates collected at the University Hospital of Zurich revealed that only one strain harbored a mutation (A2045T), which corresponded with an azithromycin-resistant phenotype on both media measured by disk diffusion, as well as by broth microdilution. Despite displaying a wild-type 23S rRNA gene, 15 out of 44 isolates presented with a resistant profile on MH and RPMI-1640 agar but showed lower MICs on RPMI-1640 than on MH medium (Figure 3).

## 3. Discussion

In this study, we showed that the azithromycin susceptibility of clinical *Enterobacteriaceae* isolates was higher on RPMI-1640 agar than on the routinely used MH agar. These findings take the previous observations, showing lower MICs in liquid tissue culture media (DMEM and RPMI-1640) than in MH broth for several bacterial species, e.g., *Stenotrophomonas*, a step further, setting the stage to expand routine AST testing in the clinical microbiology laboratory [16,17,18,21]. To the best of our knowledge, this is the first method comparison study comprising the most relevant Gram-negative bacterial species regarding antibiotic susceptibility on MH and RPMI solid and liquid media. The assumption that the growth medium has no influence on the predictive power of susceptibility testing for therapeutic effectivity is therefore put into question; and careful evaluation and revision of in vitro AST methods might be necessary for specific situations, such as infections with multidrug-resistant Gram-negative bacteria.

The observed differences in susceptibility suggest the existence of determinants influencing antibiotic susceptibility that may be active in the nutrient-rich MH agar and inactive in the more physiological and nutrient-poor RPMI-1640 agar. Ersoy et al. previously showed that the increased susceptibility of azithromycin, erythromycin and tetracycline in liquid DMEM was influenced by the presence of sodium bicarbonate in DMEM [18]. Although RPMI-1640 contains similar concentrations of sodium bicarbonate as DMEM, only the susceptibility towards azithromycin was affected in the disk diffusion assay in our study (Appendix A).

As a possible mechanism for the increased azithromycin susceptibility of *P. aeruginosa* in RPMI-1640 medium, Buyck et al. proposed an increase of the outer membrane permeability. Thereby, the macrolides are able to penetrate the membrane and, through a yet unknown mechanism, impair the expression of *oprM*, the outer membrane component of one of the main efflux systems [17,22,23]. The distinct susceptibility phenotypes we observed amongst the tested *P. aeruginosa* strains (Figure 2; Figure 3) suggest that RPMI-1640 medium induces additionally a specific response in distinct bacterial populations. Such a response might be caused by point mutations in the gene coding for Domain V of the 23S rRNA, which are known to increase resistance towards azithromycin in *P. aeruginosa* [24]. Mustafa et al. analyzed a collection of over 300 clinical *P. aeruginosa* isolates for *23S rRNA* mutations and discovered single or double mutations in 43% of the isolates with cystic fibrosis origin [21]. In contrast, no mutations were found in isolates from patients with hospital-acquired pneumonia. In our *P. aeruginosa* collection, where cystic fibrosis patients were a minority, we found only one isolate carrying a point mutation in the *23S rRNA* gene. Nevertheless, additional 15 isolates without mutation showed similar inhibition zone diameters on both media but lower MICs in RPMI-1640 medium. While resistance in a liquid medium correlated with the known mutations in Domain V of the *23S rRNA* gene, these mutations seemed to not be the sole determinants of decreased susceptibility on RPMI-1640 agar. Therefore, we assume that yet another mechanism is responsible for the medium-dependent shift in susceptibility measured by disk diffusion in our clinical isolate collection.

In a murine infection model of MDR Gram-negative bacterial infections, azithromycin was found to improve disease outcome [15,25]. Further, azithromycin was shown to reduce absolute bacterial numbers and increase specialized metabolite production of *P. aeruginosa* biofilms [26]. Thus, not only immunomodulatory effects but also impairment of planktonic and biofilm-associated bacteria may play an important role for the beneficial effects of macrolide therapy found in patients with Gram-negative sepsis or ventilator-associated pneumonia [9,11,26,27,28,29].

While several studies showed increased azithromycin MICs in tissue culture media as compared to MH in various Gram-negative species, AST by disk diffusion on tissue culture medium has not been tested so far. Comparison of the two methods revealed excellent MIC/zone correlations on RPMI-1640 agar in *E. coli*, *E. cloacae* and *P. aeruginosa*, a trend was observed in *K. pneumoniae*. These findings could pave the way for further expanding the well-established, accurate, simple, fast, and reproducible disk diffusion method from MH agar to solid RPMI-1640 to the clinical microbiology routine testing. This may broaden the potential therapeutic options in multidrug-resistant bacterial infections, eliminating major errors in forecasting therapeutic success in susceptibility reports.

In summary, we have demonstrated that the inhibition zone diameters correlated well with the MICs in both MH as well as RPMI-1640 medium. Azithromycin inhibition zone diameters significantly increased on RPMI-1640 agar when tested with the disk diffusion method, both for fully susceptible as well as multidrug-resistant Gram-negatives, including carbapenemase-producing *Enterobacteriaceae*. We took previous findings a step further by rendering them applicable in the routine setting of a clinical microbiology laboratory. Thus, we propose that for multidrug-resistant Gram-negative bacteria additional testing conditions, which more closely reflect the host environment, e.g., cultivation on RPMI-1640 agar, should be considered in routine testing in defined situations, such as assessing additional therapeutic options for multidrug-resistant Gram-negative bacteria. This may unveil additional susceptibilities to active antimicrobial substances and pave the way to clinical studies correlating standardized in vitro AST data with therapeutic outcome in vivo in order to further assess the therapeutic utility of macrolides in Gram-negative infections.

## 4. Materials and Methods

### 4.1. Bacterial Strains and Media

The azithromycin susceptibility of 28 *Escherichia coli,* 24 *Enterobacter cloacae,* 26 *Klebsiella pneumoniae* and 26 *Pseudomonas aeruginosa* isolates, collected between 2014 and 2016 at the University Hospital Zurich (USZ), was tested by disk diffusion and broth microdilution techniques. Carbapenemase-positive isolates were collected between 2013 and 2015 (14 *K. pneumoniae,* 2 *E. coli,* 1 *Proteus mirabilis,* 1 *Citrobacter koseri*, 1 *Morganella morganii)* and tested by disk diffusion. Additionally, 29 clinical *P. aeruginosa* isolates, which were collected in 4 different European study centers and had previously been tested for their susceptibility to macrolides including azithromycin by broth microdilution, [21] were now assessed by disk diffusion. Strains were stored at −80 °C in LB/20% glycerol and grown in Luria Broth Lennox (LB, Becton Dickinson, Le Pont de Claix, France) or on cation-adjusted Mueller-Hinton II (MH, BD, France) or Roswell Park Memorial Institute agar (RPMI-1640, Gibco, Paisley, UK) in a shaking incubator at 35 °C, 5% CO_2_. RPMI-1640 was prepared according to the manufacturer’s instructions (including the addition of 2 g/L NaHCO_3_), filter-sterilized, and the pH was adjusted to 7.0–7.4. RPMI-1640 agar plates consisted of 72% RPMI-1640, 1% agar (BD, France) and 3% LB. RPMI-1640 agar plates (Biomérieux, Marcy-L’Etoile, France) were used for comparison.

### 4.2. Disk Diffusion Testing

Disk diffusion testing was performed according to EUCAST guidelines using commercial MH agar square plates (Becton Dickinson, Franklin Lakes, NJ, USA) and custom-made RPMI-1640 agar plates (as described above) [30]. After placing antibiotic paper disks (azithromycin, erythromycin, clarithromycin, tobramycin, cefepime, tetracycline, tigecycline, i2a, France; for concentrations see Appendix A), plates were incubated (18 hours, 35 °C) with 5% CO_2_ to maintain a stable plate pH since RPMI-1640 was buffered with sodium bicarbonate. Inhibition zones were detected by the SirScan 2000 automatic plate reader (i2a). Two independent measurements were performed on each isolate. The quality control protocol (QC), as recommended by EUCAST for MH agar, was routinely followed also for RPMI-1640 agar to ensure standardization of the technique and quality of the agar plates [31]. Inhibition zones of the two EUCAST reference strains *P. aeruginosa* ATCC 27853 and *E. coli* ATCC 25922 matched EUCAST QC requirements on MH agar plates, confirming constant quality of our testing procedure.

### 4.3. Broth Microdilution Susceptibility Testing

The minimal inhibitory concentration (MIC) of azithromycin in liquid MH or RPMI-1640 medium was assessed by broth microdilution following the EUCAST protocol [32]. Differing from the EUCAST protocol, the plates were incubated for 20 hours at 37 °C in a shaking incubator supplemented with 5% CO_2_ to keep the pH of the RPMI-1640 medium stable. Assays were performed in duplicates.

### 4.4. Sequencing of the Gene Coding for Domain V of the 23S Ribosomal Subunit

Domain V of the *23S rRNA* gene was amplified and sequenced as described before and screened for the previously described mutations A2045G/T, A2046G/T and C2598G/T in *P. aeruginosa* strains [21].

### 4.5. Statistical Analysis

The statistical analysis was performed with either GraphPad Prism 7.04 (Mann-Whitney Test) (GraphPad Software, San Diego, CA, USA) or R version 3.5.0 (Kendall rank coefficient) (GraphPad Software, San Diego, CA, USA) software for Windows.

## Figures and Tables

**Figure 1 antibiotics-09-00218-f001:**
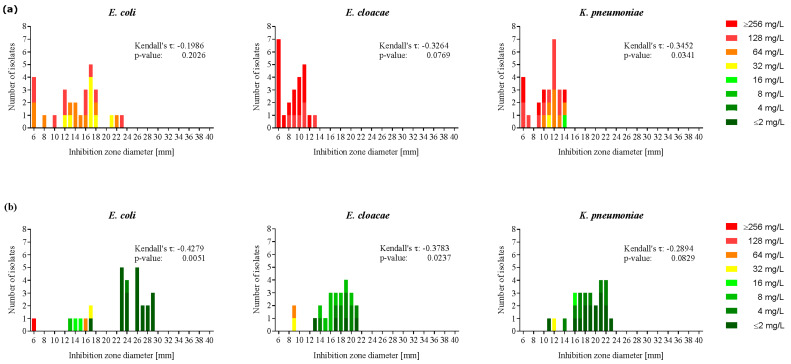
Increased azithromycin susceptibility of *Enterobacteriaceae* in RPMI-1640 medium as compared to Mueller-Hinton medium. Azithromycin susceptibility of 28 *E. coli*, 24 *E. cloacae* and 26 *K. pneumoniae* isolates in Mueller-Hinton (**a**); and RPMI-1640 (**b**); the medium was measured by disk diffusion and broth microdilution. A MIC-colored zone method was applied to correlate the inhibition zone diameter (mm) of each bacterial isolate with the corresponding MIC (mg/L). Kendall’s τ and p-value were calculated for each species and medium. Each isolate was tested in two independent experiments. Disk size: 6 mm, 15 µg azithromycin.

**Figure 2 antibiotics-09-00218-f002:**
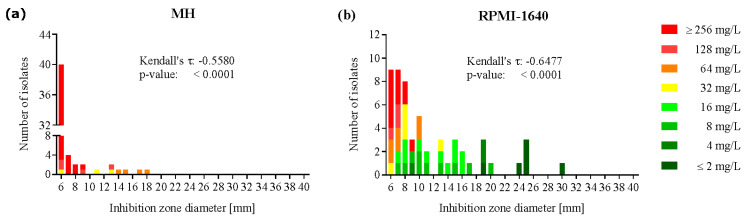
Increased azithromycin susceptibility in a subgroup of *P. aeruginosa* in RPMI-1640 medium as compared to Mueller-Hinton medium. Azithromycin susceptibility of 55 *P. aeruginosa* isolates in Mueller-Hinton (**a**); and RPMI-1640 (**b**); the medium was measured by disk diffusion and broth microdilution. MIC-zone method was applied to correlate the inhibition zone diameter [mm] of each bacterial isolate with the corresponding MIC (mg/L). Kendall’s τ and p-value were calculated for each species and medium. Each isolate was tested in two independent experiments. Disk size: 6 mm, 15 µg azithromycin.

**Figure 3 antibiotics-09-00218-f003:**
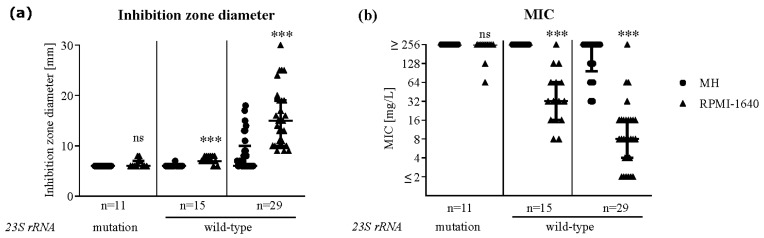
*P. aeruginosa* isolates carrying a mutation in Domain V of the *23S rRNA* gene showed a resistant profile in liquid and solid Mueller-Hinton and RPMI-1640 medium. Domain V of the *23S rRNA* gene was sequenced in 55 *P. aeruginosa* isolates and analyzed for known mutations conferring resistance to macrolides (A2045T, A2045G, A2046T, A2046G and C2598T). Isolates with a wild-type Domain V of the *23S rRNA* gene were split into two subgroups according to their resistance profile on RPMI-1640 agar. Azithromycin susceptibility was measured by disk diffusion (**a**); and broth microdilution (**b**); in Mueller-Hinton (circles) and RPMI-1640 (triangles) medium. Data show the median + interquartile range, Nonparametric Mann-Whitney Test, *** <0.0001. Disk size: 6 mm, 15 µg azithromycin.

**Table 1 antibiotics-09-00218-t001:** Median azithromycin inhibition zone diameters of *Enterobacteriaceae* and *P. aeruginosa* on Mueller–Hinton (MH) and RPMI-1640 agar.

Family	Species	*n*	MH [mm]	RPMI-1640 [mm]
	*E. coli*	28	15.5 (12–17)	24 (18.5–26.75) ^***^
*Enterobacteriaceae*	*E. cloacae*	24	9 (6–11)	17.5 (15.25–19) ^***^
	*K. pneumoniae*	26	11.5 (9–12.25)	19 (16.75–21) ^***^
*Pseudomonadaceae*	*P. aeruginosa*	55	6 (6–7)	9 (7–15) ^***^

Azithromycin inhibition zone diameters of Enterobacteriaceae (*E. coli*, *E. cloacae*, *K. pneumoniae*) and *P. aeruginosa* isolates (26 originating from the University Hospital Zurich (USZ), 29 from four different European study centers) on MH and RPMI-1640 agar. Each isolate was tested in two independent experiments. Data show median + interquartile range. Nonparametric Mann-Whitney Test, *** < 0.0001. Disk size: 6 mm, 15 µg azithromycin. *n* = number of tested clinical isolates.

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
