# Peer review of "Increased Azithromycin Susceptibility of Multidrug-Resistant Gram-Negative Bacteria on RPMI-1640 Agar Assessed by Disk Diffusion Testing"

_antibiotics, 2020, doi:10.3390/antibiotics9050218_

Round 1

Reviewer 1 Report

This manuscript appeared to be quite well written and depicted,

I am convinced by the biological results and by the conclusions  of the authors

This article appears to be of importance with respect to the criteria of quality of the journal Antibiotics and for all these reasons, I recommend this article for publication as it is.

Author Response

This manuscript appeared to be quite well written and depicted, I am convinced by the biological results and by the conclusions of the authors.

This article appears to be of importance with respect to the criteria of quality of the journal Antibiotics and for all these reasons, I recommend this article for publication as it is.

We thank Reviewer 1 for stating that our manuscript is of importance and for the recommendation for publication.

Reviewer 2 Report

1.  EUCAST, ECOFF and CLSI should be spelled out in lines 64-67.  The abbreviation EUCAST should be defined.

2.  Table 1 indicates that 55 P. aeruginosa isolates were studied, but line 239 indicates 26 isolates were used.

3.  Lines 185-186 refer to clarithromycin and erythromycin.  This should be referred to the supplementary materials.

4.  References 15, 16 & 21 use et al.  Add the full list of authors:

     15 ... Hensler ME, Burkhart MD, Pogliano J, Sakoulas G, Nizet V

     16 ... Pogliano J, Sakoulas G, Nizet V

     21 ... Van Bambeke F

Author Response

Please find the point-by-point reply in the attached file.

Reviewer 3 Report

The manuscript titled “Increased Azithromycin Susceptibility of Multi-drug Resistant Gram-negative Bacteria on RPMI-1640 Agar Assessed by Disk Diffusion Testing” addresses an important concern regarding the fight versus microorganisms. In particular, the authors show that by using a specific broth, it is possible to highlight the activity of the macrolide azithromycin against different Gram negative bacteria, thus bringing to light that using the correct broth is essential to avoid false negative. Indeed, azithromycin shows higher MIC and a smaller inhibition zone when the classical MHB is used. On the other hand, when RPMI-1640 (proposed by the authors) is used, azithromycin gets more active against different Gram-negatives species.

The manuscript is well-written and the scope of the work is rather clear.

Few imprecisions are present such as: 1) the sentence at rows 82-84 that, in my opinion, should be removed and 2) P. aeuruginosa that sometime in the text is not in italic.

However, although I think that the article addresses an essential topic within the fight against antimicrobial resistance, I believe that the data present in this manuscript are not enough to be published. Indeed, I think that the idea to careful evaluate known antibiotics in order to identify new treatments with old drugs is really successful but this idea regarding azithromycin is known. In particular, in the article “Increased Susceptibility of Pseudomonas Aeruginosa to Macrolides and Ketolides in Eukaryotic Cell Culture Media and Biological Fluids Due to Decreased Expression of oprM and Increased Outer-Membrane Permeability” (doi: 10.1093/cid/cis473) that the authors cite in their manuscript (ref. 17), the same results have been reported. Indeed, in the previous article it is demonstrated that by using the RPMI-1640 agar, azithromycin gains activity against P. aeruginosa. In addition, in that article there is also a study to understand as to why azithromycin is more active in RPMI-1640 agar that in MHB.

Therefore, the findings reported in the present manuscript add little to the information regarding the activity of azithromycin. Performing the diffusion test in addition to MIC evaluation of azithromycin on Enterobacteriaceae and P. aeruginosa strains does not involve any novelty to the field. In my opinion, these results can be considered the starting point for future in depth investigations. All the data (old and new ones) could be used for a publication.

Author Response

Please find the point-by-point reply in the attachment.

Round 2

Reviewer 3 Report

I continue to think that results from the present manuscript are not original. The proof and the explanation that Azithromycin gains activity in RPMI-1640 broth has been already reported six years ago. What the authors do, it is a linear correlation and they add data only using a different test.

However, I understand the point of the authors regarding the observation that most of the clinical microbiology laboratories are not equipped to perform broth microdilution assays but routinely can use agar-based disk diffusion testing. Therefore, from this perspective the work of the authors acquires a different value.

To conclude, I would like to explain my previous thought related to “these results can be considered the starting point for future in depth investigations. All the data (old and new ones) could be used for a publication”. I meant that the authors should have collected more innovative experiments to be added to those existing in order to obtain enough data for a publication.